# ATTENTION NEEDS TO FOCUS: A UNIFIED PERSPECTIVE ON ATTENTION ALLOCATION

## ABSTRACT

The Transformer architecture, a cornerstone of modern Large Language Models (LLMs), has achieved extraordinary success in sequence modeling, primarily due to its attention mechanism. However, despite its power, the standard attention mechanism is plagued by well-documented issues: representational collapse and attention sink. Although prior work has proposed approaches for these issues, they are often studied in isolation, obscuring their deeper connection. In this paper, we present a unified perspective, arguing that both can be traced to a common root—improper attention allocation. We identify two failure modes: 1) Attention Overload, where tokens receive comparable high weights, blurring semantic features that lead to representational collapse; 2) Attention Underload, where no token is semantically relevant, yet attention is still forced to distribute, resulting in spurious focus such as attention sink. Building on this insight, we introduce Lazy Attention, a novel mechanism designed for a more focused attention distribution. To mitigate overload, it employs positional discrimination across both heads and dimensions to sharpen token distinctions. To counteract underload, it incorporates Elastic-Softmax, a modified normalization function that relaxes the standard softmax constraint to suppress attention on irrelevant tokens. Experiments on the FineWeb-Edu corpus, evaluated across nine diverse benchmarks, demonstrate that Lazy Attention successfully mitigates attention sink and achieves competitive performance compared to both standard attention and modern architectures, while reaching up to 59.58% attention sparsity.

## 1 INTRODUCTION

The remarkable success of Large Language Models (LLMs) across a diverse range of tasks, from text generation (Radford et al., 2019) to complex reasoning (Wei et al., 2022), hinges on the Transformer architecture's core innovation: the self-attention mechanism. By computing pairwise similarity scores between queries and keys, self-attention enables a model to dynamically construct context-aware representations. The underlying principle is to assign greater weight to semantically relevant tokens and lesser weight to irrelevant ones, thereby enabling the model to capture meaningful dependencies across tokens in the sequence.

However, empirical evidence shows that the standard attention mechanism often deviates from this ideal, giving rise to two widely observed phenomena: representational collapse (Barbero et al., 2024) and attention sink (Xiao et al., 2023). Representational collapse occurs in relatively long-context scenarios, where the attention mechanism tries to aggregate information from too many tokens, leading to over-squashing and ultimately indistinguishable final representations. Conversely, attention sink describes that the initial tokens disproportionately attract high attention weights and serve as anchors to stabilize the attention distribution even though they carry little semantic significance (Gu et al., 2024). Both phenomena degrade model performance and hinder efficient deployment, suggesting a deeper structural deficiency in current attention design.

Existing works address these problems separately. For the collapse issue, common approaches increase model dimensionality or numerical precision, which alleviates representation loss but at the cost of significantly higher training and inference overhead. For the sink phenomenon, prevailing solutions adapt to the bias by explicitly preserving a sink token (Xiao et al., 2025) or sink bias (OpenAI et al., 2025), which reduces model flexibility, or by removing normalization, which often comes

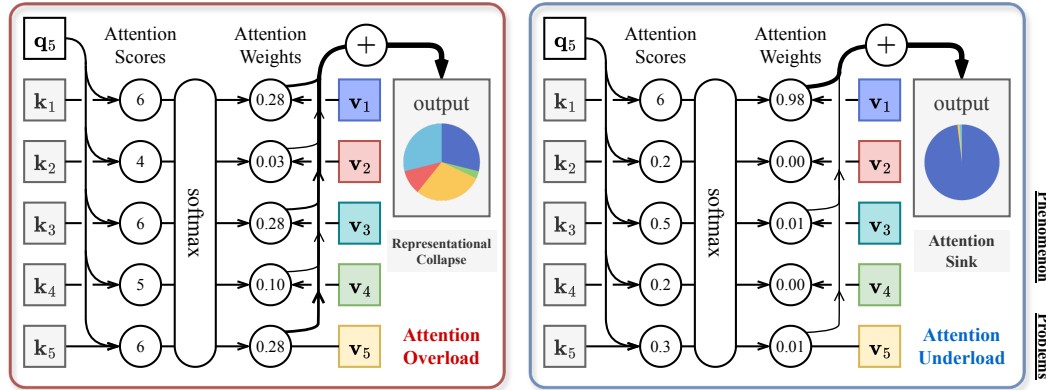

Figure 1: Overview of two attention failure modes. Attention Overload: broadly assigned high weights blur important patterns. Attention Underload: uniformly low relevance gets normalized.

at the expense of performance stability. Although such approaches mitigate individual issues, they are inherently fragmented and fail to handle both problems within a unified framework.

In this paper, we provide a unified perspective on the above problems: *the core limitation of self-attention lies in its improper attention allocation*, manifesting in two opposite extremes (Figure 1). **Attention Overload** occurs when dense contexts spread attention too broadly across tokens, causing semantic features to be averaged away and resulting in indistinguishable representations (Sun et al., 2025). **Attention Underload** when a token has little relevance to the previous context, yet softmax normalization requires the attention weights to sum to one, leading to the sink phenomenon. Together, these two failure modes highlight the need for an attention mechanism that allocates focus more selectively, amplifying the attention weight of informative tokens while ignoring irrelevant ones.

Therefore, we propose Lazy Attention, which integrates two complementary mechanisms to improve focus allocation: 1) **Positional Discrimination**: This mechanism combines RoPE with learnable attention biases to enhance positional feature differentiation across attention heads and head dimensions. By reducing the confusion of token representations it helps to alleviate attention overload. 2) **Elastic-Softmax**: This variant introduces an offset to the attention weights, relaxing the strict normalization constraint of standard softmax. It enables the model to assign zero weights to irrelevant tokens in underload situations and eliminates the attention sink problem. Together, these mechanisms encourage sparse and focused attention that is better aligned with semantic relevance, thereby improving the effectiveness of language modeling.

Our contributions can be summarized as follows:

- We provide a unified perspective that characterizes attention overload and underload as two fundamental failure modes of attention allocation.
- We propose Lazy Attention, which (i) combines RoPE with learnable attention bias to enhance positional discrimination across heads and dimensions to alleviate attention overload, and (ii) uses Elastic-Softmax to suppress attention sink by filtering out negligible weights.
- Extensive experiments across nine benchmarks show that Lazy Attention mitigates attention sink, achieves an average sparsity of 59.58% in the attention weights, and improves language modeling performance, confirming the importance of enhanced focus in attention design.

## 2 PRELIMINARIES

### 2.1 TRANSFORMER ARCHITECTURE

A Transformer (Vaswani et al., 2017) is composed of $L$ stacked layers, each consisting of a multi-head attention (MHA) module and a feed-forward network (FFN), both equipped with residual connections. Given an input sequence $\boldsymbol{X}^l = [\boldsymbol{x}_1, \boldsymbol{x}_2, ..., \boldsymbol{x}_n] \in \mathbb{R}^{n \times d}$ at layer $l$, where $n$ is the sequence



(a) Sink position with $n$-th replaced by a fix token during pre-training

(b) Transformers with standard attentions and the sink phenomenon

(c) Transformers with sliding window attention (SWA)

Figure 2: Analysis of the location of the attention sink and the characteristics of the sink token.

length and $d$ the hidden dimension, the layer update can be written as:

$$\hat{X}^l = X^l + \text{MHA}(\text{LN}(X^l)), \quad X^{l+1} = \hat{X}^l + \text{FFN}(\text{LN}(\hat{X}^l)), \tag{1}$$

where $\text{LN}(\cdot)$ denotes layer normalization. The MHA mechanism enables the model to capture information from multiple representation subspaces simultaneously. For each attention head $h \in [H]$, the queries, keys, and values are computed as: $Q_h = X^l W_h^Q$, $K_h = X^l W_h^K$, $V_h = X^l W_h^V$, where $W_h^Q, W_h^K, W_h^V \in \mathbb{R}^{d \times d_h}$ are projection matrices. Finally, the attention operation is:

$$\text{Attention}(Q_h, K_h, V_h) = \text{softmax}\left(Q_h K_h^\top / \sqrt{d_h}\right) V_h, \tag{2}$$

where $d_h = d/H$. The attention weight, $\text{softmax}(Q_h K_h^\top / \sqrt{d_h}) \in \mathbb{R}^{n \times n}$, forms a probability distribution over the sequence for each query position.

### 2.2 POSITION ENCODING

Since self-attention is permutation-invariant, positional information must be incorporated into the model. Transformer (Vaswani et al., 2017) introduced sinusoidal functions to create fixed positional encodings, which often referred to as **absolute position encodings (APE)**, are defined as $P_{(i,2t)} = \sin(i/10000^{2t/d})$ and $P_{(i,2t+1)} = \cos(i/10000^{2t/d})$. where $i$ is the token position, $t$ is the dimension and $d$ is the hidden dimension size. Subsequent works like GPT (Radford et al., 2019) adopted learnable position embeddings provided by model parameters. In both variants, position embeddings $P \in \mathbb{R}^{n \times d}$ are added to token embeddings: $\text{TokenEmbed}(X) + P$ and input to the first Transformer layer. However, APEs have poor length generalization beyond training sequences.

To address the limitations of APE, **relative position encoding (RPE)** was introduced to model the pairwise distance between tokens. Instead of adding embeddings, RPEs typically modify the attention mechanism itself, especially the attention scores. Influential methods include Transformer-XL (Dai et al., 2019), which uses a granular attention formula that disentangles content and relative position, and ALiBi (Press et al., 2022), which simply adds a static bias (e.g., $-m \cdot |i - j|$) to the attention scores. More recently, **rotary position embedding (RoPE)** (Su et al., 2023) has become adopted in LLMs. It encodes relative position by applying rotational transformations, concisely represented as $q_i' = R_i q_i$, $k_i' = R_i k_i$, where $R_i$ is a block-diagonal rotation matrix, allowing attention scores to reflect relative positions. More related works are detailed in Appendix B.1

In summary, position encoding methods have evolved from absolute to relative and rotary designs, reflecting a trend toward capturing richer and more flexible positional dependencies.

## 3 RETHINKING ATTENTION SINK AND POSITION ENCODING

### 3.1 THE NATURE OF ATTENTION SINK

Attention sink refers to the phenomenon that initial tokens attract a large amount of attention during inference (Xiao et al., 2023; Han et al., 2024). Related discussions are provided in Appendix B.2.

Following Gu et al. (2024), we probe the location of the attention sink by inserting a fixed [Mask] at a chosen position during pre-training. As shown in Figure 2a, when the fixed token is close to the

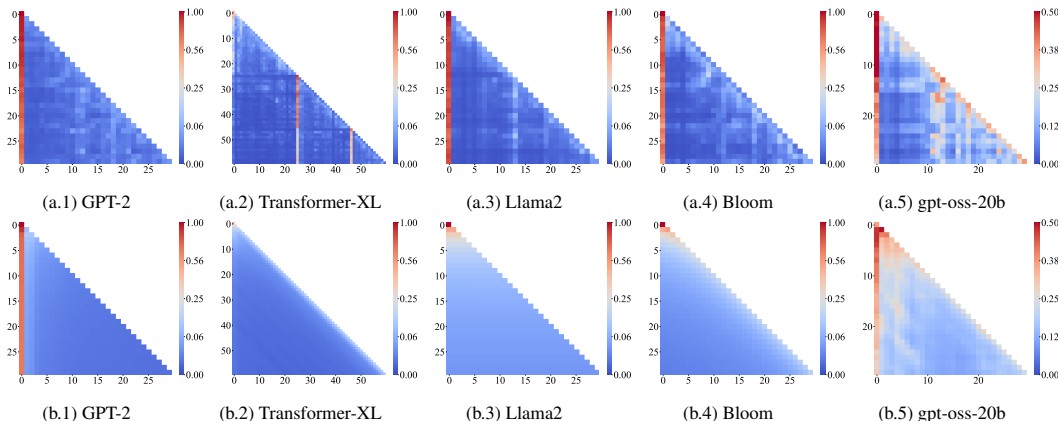

Figure 3: Comparison of attention weight patterns in five LLMs employing different position encodings. The models are evaluated with two types of input: (a) natural text and (b) repeated tokens. Under repeated tokens, the sink pattern of attention weights changes significantly.

first token, it absorbs the attention from the first position. However, when the fixed token is placed more than 16 tokens away from the starting position, the first token becomes the sink token again.

Meanwhile, Chi et al. (2023) pointed out that the variance of hidden states in Transformers contains latent positional information even without positional embeddings. This is consistent with the comparison of Figure 2b and 2c. When attention sink exists, the variance of the sink token's hidden states is much greater than that of other tokens. Similarly, the variance of the sink token's value in the attention calculation is much smaller than that of other tokens, as detailed in Appendix G.

> **Takeaway 1**: The attention sink is semantic-agnostic. It distinguishes the sink token, predominantly the first token, via the variance of the value vectors ($V$) and the hidden states.

## 3.2 THE ROLE OF POSITION ENCODING

Building on the above findings, we next examine the role of position encoding in shaping attention behavior. By feeding sequences of repeated tokens (e.g., "the the the..."), we eliminate semantic and variance differences in token embeddings, which help distinguish the sink token. Figure 3 visualizes attention weights across five representative LLMs with distinct position encodings.

With APE (e.g., GPT-2), the sink phenomenon persists even under repeated inputs (Figure 3b.1). In contrast, LLMs adopting RPE—such as Llama2 (Touvron et al., 2023) with RoPE and Bloom (Scao et al., 2022) with ALiBi—lose their sink behavior entirely. Their attention distributions collapse into uniform patterns modulated by long-term decays of position encodings. This indicates that RoPE, ALiBi, and similar RPE methods fundamentally provide position-sensitive attention weights rather than embed positional information into hidden states that can propagate across layers.

Two models exhibit particularly distinctive behaviors. Transformer-XL (Figure 3a.2) shows sinks not at the first token but at punctuation and other special tokens. This arises from its block-recurrent training scheme. In contrast, GPT-OSS preserves its sink even under repeated-token inputs. This is because GPT-OSS adds learnable biases to the softmax denominator, functioning as virtual sink tokens. These biases induce subtle differences in attention weights even among identical tokens, thereby allowing the model to retain positional distinctions.

> **Takeaway 2**: Relative position encoding shapes attention weight distributions rather than directly modifying token embeddings, thereby providing richer position-based discrimination and strengthening the model's capacity.

### 3.3 ATTENTION OVERLOAD AND UNDERLOAD

*Attention overload* arises when many tokens obtain considerable weights, forcing excessive features to be compressed into a single token and leading to collapse. RPEs induce differences in attention weights across positions, which make representations more distinguishable and alleviate this issue.

On the contrary, *attention underload* manifests as attention sink, where softmax forces attention weights to form a probability distribution that sums to one, reducing flexibility when no token is truly relevant. Specifically, given the attention scores $s_{ij}^{(h)}$ between the $i$-th query and the $j$-th key under head $h$, the attention weight is defined as:

$$\alpha_{ij}^{(h)} = \text{Softmax}(s_{ij}^{(h)}) = \frac{\exp(s_{ij}^{(h)})}{\sum_{k=1}^{i} \exp(s_{ik}^{(h)})}, \; j \leq i. \tag{3}$$

A fundamental limitation of softmax is that attention weights must always sum to one. When no token is truly relevant, the distribution has no meaningful place to settle, and the model designates the first token as a sink, where the variance of its representation is reduced, allowing it to act as a stabilizer with little influence on other features.

The above analysis reveals distinctive characteristics underlying the two failure modes of attention, which in turn suggest principles for their mitigation.

## 4 METHODOLOGY

In this section, we introduce **Lazy Attention**, which combines: 1) **Positional Discrimination** across heads and dimensions to overcome attention overload; with 2) **Elastic-Softmax** to filter out negligible attention weights to deal with attention underload. Together, they make attention focused while preserving performance and enhancing extrapolation ability.

### 4.1 POSITIONAL DISCRIMINATION

Attention overload arises when a query attends too broadly to dense contexts, forcing it to compress many heterogeneous features into a single representation. This multiplexing exceeds the representational capacity of a token and blurs semantic distinctions. To alleviate overload, we leverage positional information to guide attention weights, so that token representations remain more distinguishable even in dense contexts. We therefore propose a unified positional encoding scheme that integrates both head-wise and dimension-wise positional discrimination.

Concretely, we incorporate two complementary components into the attention score computation:

- RoPE: provides *dimension-wise* discrimination by rotating query and key vectors at different frequencies $\theta_d = B^{-2d/D}$ across head dimensions, where $B$ denotes the RoPE base. This encodes relative positions between tokens into the scaled dot-product computation. In our design, we adopt a larger $B$ to mitigate the inherent long-range decay of RoPE.
- Learnable Attention Biases: introduces *head-wise* discrimination by assigning each attention head a fixed linear bias $-m_h \cdot |i - j|$ that decays with token distance.

Unlike the original ALiBi, which enforces a uniform, static slope for long-range decay, our method allows head-specific and distance-specific biases to be learned. Overall, for each attention head $h$ and relative distance $|i - j|$, we compute:

$$s_{ij}^{(h)} = \frac{\left(\boldsymbol{R}_i q_i^{(h)}\right)^\top \left(\boldsymbol{R}_j k_j^{(h)}\right)}{\sqrt{d}} - b_{|i-j|}^{(h)}, \tag{4}$$

where $\boldsymbol{R}_i$ and $\boldsymbol{R}_j$ are the RoPE rotation matrices applied to the $i$-th query and $j$-th key under head $h$, and $b_{|i-j|}^{(h)}$ is a learnable bias shared across all queries at distance $|i - j|$ for head $h$. The learnable bias avoids the excessive long-term decay of ALiBi, where distant tokens are overly suppressed. Both RPE methods enhance positional discrimination across attention dimensions and heads.

### 4.2 ELASTIC-SOFTMAX

To address attention underload, we focus on the limitation of softmax normalization, which forces attention weights to form a probability distribution that sums to one. When no token is truly relevant, this constraint leads to unnecessary allocations and reduces the flexibility of attention. Existing work either adapts to the sink by reserving sink tokens or alters normalization at the cost of stability.

Motivated by these drawbacks, we introduce **Elastic-Softmax**, a lightweight modification that applies a simple filter after the standard softmax. This design preserves the benefits of normalization while enabling the model to suppress irrelevant tokens. Specifically, for each attention head $h$, the attention weight is calculated as:

$$\alpha_{ij}^{(h)} = \text{Elastic-Softmax}(s_{ij}^{(h)}) = \text{ReLU}\left(\text{Softmax}(s_{ij}^{(h)}) - \frac{\tau^{(h)}}{i}\right), \ \tau_0^{(h)} = 1, \qquad (5)$$

where $\tau^{(h)}$ is a learnable head-specific offset, initialized to $1$ and divided evenly across the $i$ attended tokens. ReLU (Agarap, 2018) ensures all weights remain non-negative.

Elastic-Softmax eliminates insignificant attention weights, resulting in sparse attention that mitigates attention sink and emphasizes informative tokens. For instance, if no specific token stands out as a focus of attention, attention weights are offset to zero when $\tau^{(h)} = 1$. Moreover, the formulation is fully compatible with FlashAttention (Dao et al., 2022), as detailed in Appendix F.

## 5 EXPERIMENTS

### 5.1 EXPERIMENT SETTINGS

**Datasets.** LLMs are trained on FineWeb-Edu (Lozhkov et al., 2024) 10BT and 100BT; 15B/30B are taken sequentially from shuffled 100BT, and ablations use full 10BT (see Appendix C.1 for details).

**Baselines.** We benchmark our method against: (i) Transformer variants: Transformer++ (Touvron et al., 2023) and Sigmoid Attention (Ramapuram et al., 2025); (ii) Recurrent architectures: Ret-Net (Sun et al., 2023a), Mamba2 (Dao & Gu, 2024), TTT (Sun et al., 2024), Gated DeltaNet (Yang et al., 2024), and Titans (Behrouz et al., 2024). (iii) Streaming inference: StreamingLLM (Xiao et al., 2023).

**Implementation details.** We follow the training setup of prior works (Gated DeltaNet and Titans): Llama 2 tokenizer (32k vocab), sequence length 4096, and a global batch size of 0.5M tokens. Peak LR is 4e-4 with a cosine scheduler; warmup is 1024 steps for 10B/15B and 2048 for 30B. Ablations use context 512. Lazy Attention is initialized with $\tau = 1.0$. More details are in Appendix C.2.

**Evaluation metrics.** We report perplexity (ppl), accuracy (acc), normalized accuracy (acc_n; chance-adjusted), attention density (density; mean attention mass on non-first tokens), and sink ratio (sink; mean attention mass on the first token). See Appendix C.3 for formal definitions.

### 5.2 OVERALL PERFORMANCE

As illustrated in Table 1, Lazy Attention outperforms all competing architectures at the 340M scale and remains competitive when scaled to 760M.

The "-Positional" variant encodes position with plain RoPE. Compared with this baseline, the full model achieves consistently higher accuracy, particularly on longer-context tasks such as *Wino-Grande* and *BoolQ*. This shows that combining dimension-wise rotations with head-wise biases encodes positions more effectively and generalizes beyond training length. This "-Elastic" variant removes Elastic-Softmax. Sometimes this gives slightly higher averages, but the full model strikes a better balance: it retains competitive accuracy while benefiting from substantially sparser attention. Elastic-Softmax further releases over **59.58%** of weights and removes the sink(see Section 5.6).

### 5.3 WHAT LONG-TERM DECAY DOES ATTENTION REALLY NEED?

In this section, we visualize the learned distance-dependent bias $b_{|i-j|}^{(h)}$ across layers and heads, as shown in Figure 4. The bias patterns vary across transformer layers. The **first** layers display similar

Table 1: Overall comparison of Lazy Attention and other LLM architectures on eight common-sense reasoning tasks. Bold values represent optimal performance, while second-best values are underlined. "**\***" indicates the statistically significant improvements (i.e., two-sided t-test with $p < 0.05$) over the best baseline. ↑: higher is better. ↓: lower is better.

| Model | Wiki. ppl ↓ | LMB. ppl ↓ | LMB. acc ↑ | PIQA acc ↑ | Hella. acc_n ↑ | Wino. acc ↑ | ARC-e acc ↑ | ARC-c acc_n ↑ | SIQA acc ↑ | BoolQ acc ↑ | Avg. ↑ |
|---|---|---|---|---|---|---|---|---|---|---|---|
| | | | | 340M params / 15B tokens | | | | | | | |
| Transformer++ | 25.76 | 38.02 | 33.28 | 66.65 | 39.86 | 53.51 | 58.33 | 26.88 | 38.89 | 60.95 | 47.29 |
| StreamingLLM | 35.52 | 38.02 | 33.30 | 66.43 | 39.77 | 53.35 | 58.38 | 26.88 | 39.10 | 61.01 | 47.28 |
| Sigmoid Attention | 26.65 | 36.77 | 32.80 | 66.87 | 39.38 | 53.43 | 56.90 | 27.39 | 37.62 | 58.44 | 46.60 |
| Gated DeltaNet | 24.90 | 32.35 | 33.13 | 66.43 | 41.17 | 54.06 | 58.59 | **29.18** | **39.87** | 58.99 | 47.68 |
| Titans | 25.07 | **28.72** | **36.71** | 64.88 | 40.56 | 52.49 | 57.72 | 28.16 | 39.75 | 60.01 | 47.54 |
| Lazy Attention | 25.32 | 31.84 | 35.28 | **66.92** | 41.58 | 52.25 | **60.31** | 27.56 | 38.95 | 61.22 | 48.01 |
| - Positional | 25.58 | 33.06 | 34.02 | 66.59 | 41.34 | 53.43 | 60.02 | 28.16 | 39.20 | 60.40 | 47.89 |
| - Elastic | **24.59** | 31.56 | 34.78 | **66.92** | **42.11** | **54.30** | 59.43 | 28.67 | 39.36 | **61.59** | **48.39\*** |
| | | | | 760M params / 30B tokens | | | | | | | |
| Transformer++ | 25.21 | 27.64 | 35.78 | 66.92 | 42.19 | 51.95 | 60.38 | 32.46 | 39.51 | 60.37 | 48.69 |
| RetNet | 26.08 | 24.45 | 34.51 | 67.19 | 41.63 | 52.09 | 63.17 | 32.78 | 38.36 | 57.92 | 48.46 |
| Mamba2 | 22.94 | 28.37 | 33.54 | 67.90 | 42.71 | 49.77 | 63.48 | 31.09 | 40.06 | 58.15 | 48.34 |
| DeltaNet | 24.37 | 24.60 | 37.06 | 66.93 | 41.98 | 50.65 | 64.87 | 31.39 | 39.88 | 59.02 | 48.97 |
| TTT | 24.17 | 23.51 | 34.74 | 67.25 | 43.92 | 50.99 | 64.53 | 33.81 | 40.16 | 59.58 | 47.32 |
| Gated DeltaNet | 21.18 | 22.09 | 35.54 | 68.01 | 44.95 | 50.73 | **66.87** | 33.09 | 39.21 | 59.14 | 49.69 |
| Titans | **20.04** | 21.96 | 37.40 | 69.28 | **48.46** | 52.27 | 66.31 | **35.84** | 40.13 | **62.76** | **51.56** |
| Lazy Attention | 24.08 | **20.92** | **39.72** | 69.59 | 48.34 | **53.83** | 65.40 | 33.53 | **40.43** | 61.42 | 51.53 |

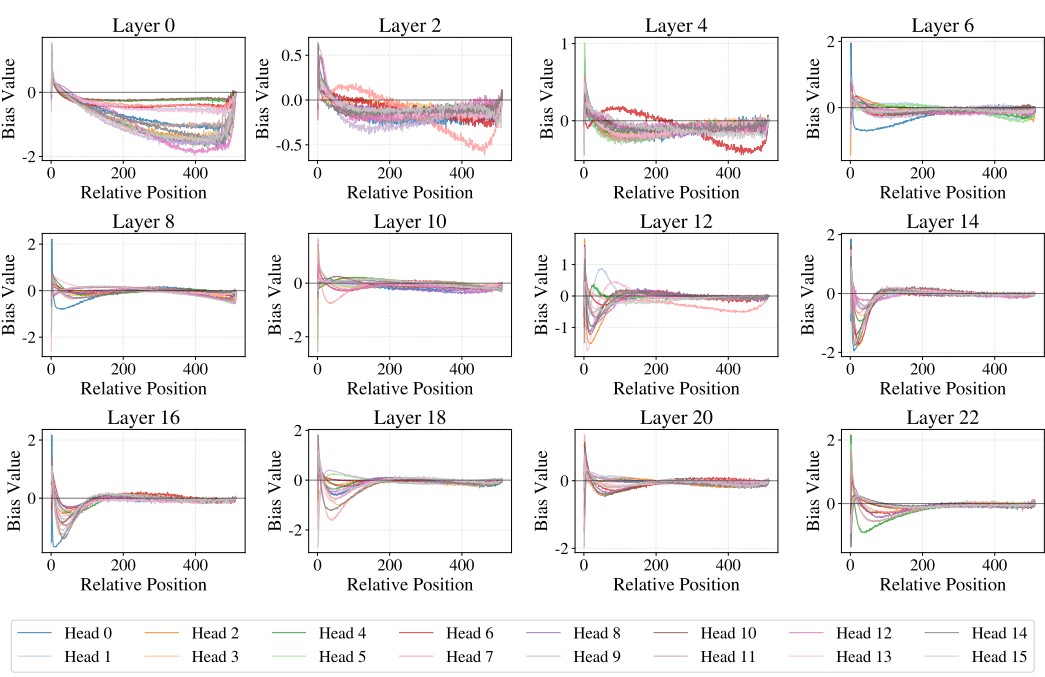

Figure 4: Learned position-dependent attention biases across different layers and attention heads.

linear decay with distance with ALiBi, which helps the model focus on local context during initial feature extraction. **Middle** layers (Layers 2-10) keep this general decay pattern but with much smaller values, about 50% lower than the first layer, suggesting a gradual relaxation of locality constraints as representations become more abstract. Notably, **upper** layers (Layers 10-24) no longer penalize distant positions, allowing the model to access global context when needed.

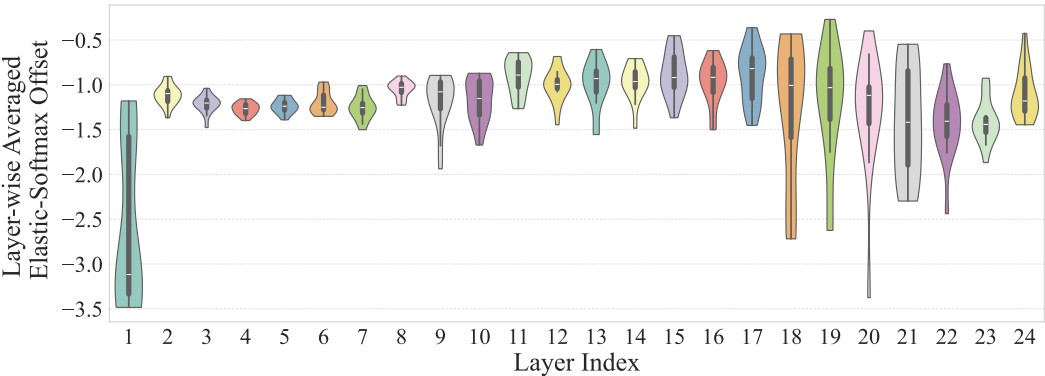

Figure 5: Distribution of learned Elastic-Softmax offsets across layers, with violin plots showing variation across heads and white dots denoting per-layer means.

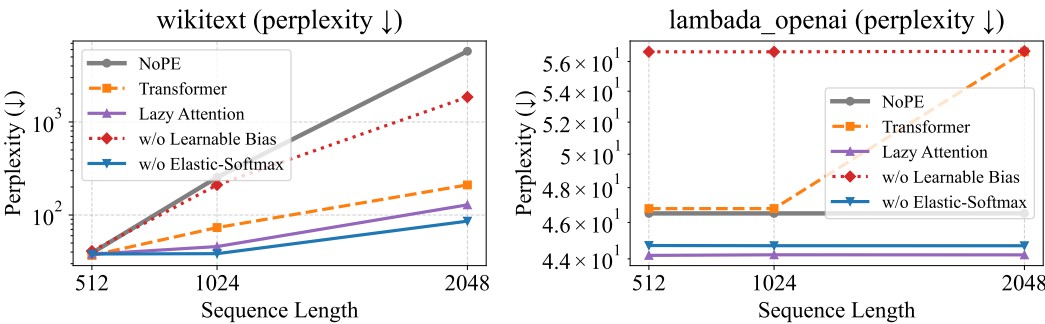

Figure 6: Perplexity comparison on WikiText and LAMBADA (OpenAI) across different sequence lengths (512, 1024, 2048). The models are pre-trained with a context length of 512.

**Inhibition of Return (IOR).** The upper layers display a non-monotonic bias characterized by three unique patterns: (i) positive bias for immediate neighbors, followed by a sharp descent to negative values (1-10), (ii) gradual recovery (10-100) to zero, and (iii) stabilization near zero for all distant positions. This structure resembles inhibition-of-return (IOR) (Klein & Ivanoff, 2000) in human visual attention, where recently attended locations are temporarily suppressed to promote exploration. This spontaneous emergence indicates that resolving overload by suppressing redundant mid-range attention while preserving access to long-range dependencies benefits language modeling.

## 5.4 ADAPTIVE FOCUSED ATTENTION

Figure 5 visualizes the layer-wise distribution of Elastic-Softmax offsets. These offsets complement the learned distance-dependent biases by providing head-specific filtering. The offset distributions also reveal three distinct stages across network depth. The **first** layer uses exceptionally large negative offsets, indicating aggressive filtering of weak attention weights during initial layer processing. Layers **2-17** maintain stable offsets around $-1.0$, suggesting consistent background noise removal throughout the main computation phase. This stability aligns with our goal of eliminating attention sink artifacts without excessively pruning informative connections. **Upper** layers ($\geq 18$) exhibit increased variance, indicating the concentration of attention on the most relevant tokens before final prediction. The learned offset progression follows a clear computational logic: aggressive initial sparsification, stable intermediate processing, and final refinement. This elastic mechanism enables the model to dynamically adjust focus by model depth and token relevance, thereby eliminating underload and removing the sink phenomenon.

Table 2: Comparison of different variants of Elastic-Softmax offset formulations. We report validation loss (↓), average performance across benchmarks (↑), density of attention weights (↓), and sink ratio (↓). Bold values indicate the best performance within each metric.

| Model | Loss ↓ | Perf. ↑ | Density ↓ | Sink ↓ |
|---|---|---|---|---|
| $\text{softmax}(QK^T/\sqrt{d})$ | **2.62** | 45.48 | 94.53 | 5.46 |
| $\text{ReLU}(\text{softmax}(QK^T/\sqrt{d}) - \tau_h), \tau_h^{(0)} = 0$ | 2.75 | 43.83 | **15.13** | 1.66 |
| $\text{ReLU}(\text{softmax}(QK^T/\sqrt{d}) - \tau_h/\text{seq\_len}), \tau_h^{(0)} = 0$ | 2.65 | 45.29 | 37.30 | 4.15 |
| $\text{ReLU}(\text{softmax}(QK^T/\sqrt{d}) - \tau_h/\text{seq\_len}), \tau_h^{(0)} = 1$ | 2.64 | **45.70** | 40.24 | **0.18** |
| $\text{ReLU}(\text{softmax}(QK^T/\sqrt{d}) - 1/\text{seq\_len})$ | 2.67 | 45.40 | 43.16 | 0.54 |

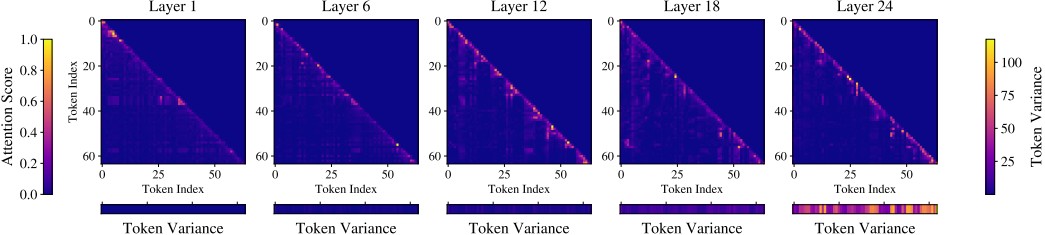

Figure 7: Visualization of the sink-free attention weight of Lazy Attention.

## 5.5 RESOLUTION OF OVERLOAD: LENGTH EXTRAPOLATION

To validate that our architecture mitigates attention overload, we evaluate length extrapolation by measuring perplexity at test lengths 512, 1024, and 2048 for models pretrained with a context length of 512 (Figure 6). On both datasets, Lazy Attention exhibits substantially smaller degradation as sequence length increases compared to the standard Transformer, indicating improved robustness to long contexts. Ablations show that removing position-aware biases severely hurts long-length performance. Overall, these results show that the full Lazy Attention design offers the best trade-off between perplexity and sparsity for extended contexts, and that positional discrimination is crucial for reducing attention overload and postponing the representational collapse.

## 5.6 RESOLUTION OF UNDERLOAD: FOCUSED ATTENTION

We also tested several variants of the Elastic-Softmax offset designs in Table 2. Fixed offsets, whether independent of sequence length ($\tau_h$) or scaled as $1/\text{seq\_len}$, lead to performance drops despite reducing sink or increasing sparsity. In contrast, initializing $\tau_h^{(0)} = 1$ and adapting it as $\tau_h/\text{seq\_len}$ provides the best trade-off, preserving accuracy while maintaining focused attention. Meanwhile, Figure 7 shows that Lazy Attention substantially alleviates attention sink and reduces excessive variance in the initial tokens, suggesting that our design addresses the attention underload issue by preventing surplus attention from being assigned to irrelevant tokens.

## 6 CONCLUSION

In this paper, we presented Lazy Attention, a unified design to improve focus allocation in self-attention. Our analysis identified two attention extremes: overload, which leads to representational collapse, and underload, which causes attention sink. Lazy Attention addresses these problems by enhancing positional discrimination and relaxing strict normalization to suppress irrelevant tokens. Experiments across diverse benchmarks confirmed that it eliminates attention sink and improves general performance while inducing substantial sparsity in attention weights. These findings suggest a promising direction for developing more efficient implementations in future LLMs.

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

## A  LLM Usage Disclosure

This document was created with assistance from AI tools for language polishing. The content has been reviewed and edited by human authors. For more information on the extent and nature of AI usage, please contact the author.

## B  Related Works

### B.1  Position Encoding

Position encoding is essential in Transformer models since self-attention itself is permutation-invariant. Early studies mainly adopted absolute position embeddings (APE). The original Transformer used fixed sinusoidal functions, and GPT-2 (Radford et al., 2019) later replaced them with trainable vectors fused with token representations. Although these approaches worked well for sequences within the training length, they struggled to generalize once the context became longer.

Relative position encoding (RPE) offered a more flexible alternative. Transformer-XL (Dai et al., 2019) introduced learnable biases based on token distance, which allowed recurrent processing between segments. ALiBi (Press et al., 2022) simplified the idea by applying fixed linear biases that decay with distance, making it possible to extrapolate to much longer inputs. These methods shifted attention from absolute indices to relative distances, laying the groundwork for position encodings that are better suited to scaling Transformers toward long and variable contexts.

Rotary Position Embeddings (RoPE) (Su et al., 2023) marked another step forward by encoding relative positions as rotations applied to the query and key vectors. RoPE has since become standard in many LLMs. Building on this, several extensions have been proposed. For example, YaRN rescales the rotation angles to push LLaMA models to million-token contexts with limited retraining (Peng et al., 2024). LongRoPE2 searches for scaling factors through evolutionary methods to balance precision in both short and long ranges (Shang et al., 2025). XPOS combines blockwise causal attention with midrange relative biases to improve interpolation stability (Sun et al., 2023b).

Other designs, such as Fourier Position Embedding (FoPE) (Hua et al., 2024), emphasize frequency filtering to retain periodic structure and help models remain stable when context length grows.

In short, position encoding has progressed from absolute sinusoidal forms to relative, rotational, and frequency-based strategies. Recent work pays as much attention to the balance of decay, frequency preservation, and interpolation stability as it does to simply extending the window length.

### B.2 ATTENTION SINK

Attention sink refers to the phenomenon that initial tokens attract a large amount of attention during inference (Xiao et al., 2023; Han et al., 2024). Subsequent analysis links this to normalization, which forces LLMs to assign attention even when no informative value (Gu et al., 2024).

To mitigate potential issues associated with this phenomenon, various solutions have been developed. Most models try to adapt this pattern by leaving the first token as the sink token and always keeping it during inference (Xiao et al., 2025; Jiang et al., 2024). Similarly, GPT-OSS (OpenAI et al., 2025) adds a learnable parameter to the denominator of the softmax function, which is equal to create a "hidden" sink token with constant attraction, thereby alleviating the sink issue without requiring a physical placeholder token. Sigmoid attention (Ramapuram et al., 2025) can completely avoid attention sink while sacrificing model performance and training stability.

### B.3 SPARSE AND EFFICIENT ATTENTION

Another challenge comes from the quadratic cost of self-attention. Fixed-pattern sparsity, exemplified by Longformer (Beltagy et al., 2020), adopts sliding windows with global tokens, which improves efficiency but reduces flexibility. Linformer (Katharopoulos et al., 2020) takes a different route by approximating attention with low-rank projections, reducing the cost to linear time.

More recent work has looked at dynamic and normalization based mechanisms. Among them, StreamingLLM (Xiao et al., 2023) introduced the notion of an "attention sink," caching only a few initial tokens to enable million-token inference even for models trained on short windows. Sigmoid Attention (Ramapuram et al., 2025) implemented a sigmoid variant of attention in a hardware-friendly way, achieving more than 15% faster inference without sacrificing accuracy. These approaches illustrate how relatively small design changes can make long-context inference practical.

A complementary direction is cache optimization. DuoAttention (Xiao et al., 2025) splits heads into retrieval and streaming roles, keeping full key–value caches only for retrieval to reduce memory. RetroAttention (Choi et al., 2025) refreshes caches with new entries to improve coverage of early tokens. FlexPrefill (Lai et al., 2025) adapts the prefill stage by selecting query-aware sparse indices, balancing accuracy and efficiency during initialization. Together, these methods show how careful memory management can extend usable context length without prohibitive compute costs.

Research on sparse and efficient attention has moved from fixed patterns to adaptive and cache-aware designs. Instead of focusing solely on reducing asymptotic complexity, recent methods also aim to balance speed, memory footprint, and robustness in real-world deployment.

## C DETAILED EXPERIMENT SETTINGS FOR REPRODUCIBILITY

### C.1 DATASETS

All experiments use subsets of FineWeb-Edu (Lozhkov et al., 2024). We train on 10B-token (10BT) and 100B-token (100BT) pools; preprocessing applies a global shuffle (seed=42). The 15B and 30B budgets are obtained deterministically by taking the first 15B / 30B tokens from the start of the shuffled 100BT subset. Ablation runs use the full 10BT pool, which is shuffled once in preprocessing and then used with a fixed order during training. For all model sizes and runs the training data and its order are identical (the same shuffled are used), ensuring data-level consistency across experiments.

## C.2 IMPLEMENTATION DETAILS

We align our training recipe with prior work (Gated DeltaNet, Titans). All runs use the Llama 2 tokenizer (32k vocab), sequence length 4096, and a global batch of 0.5M tokens. Optimization uses adamw_torch_fused with $(\beta_1, \beta_2) = (0.9, 0.95)$, weight decay 0.01, peak LR $4 \times 10^{-4}$ and a cosine_with_min_lr schedule; warmup = 1024 steps for 10B/15B runs and 2048 steps for 30B. We initialize $\tau = 1.0$. Ablation runs use the 10B subset with context length 512. For transparency and reproducibility: the 340M results are from models we trained under the above protocol; 730M (Titans) numbers are cited from the original papers and marked as reported.

All models and baselines are implemented on top of the FLA (Yang & Zhang, 2024) and Flame (Zhang & Yang, 2025) stack, which provides Triton-based efficient-attention kernels and standard LM training utilities. For sigmoid-attention experiments we replace the FlashAttention kernel with the authors' Flash-Sigmoid implementation integrated into the same stack.

## C.3 EVALUATION METRICS

**Normalized accuracy (acc_n).** Normalized accuracy adjusts raw accuracy for dataset-specific chance performance. We compute a chance baseline for each dataset (e.g., majority class or random-chance) and rescale accuracy relative to this baseline so that scores are comparable across datasets of different difficulty.

**Attention density (density).** Attention density measures how much attention weight is assigned to non-first tokens. Concretely, we compute the mean mass assigned to keys other than the first token, then average these values of all queries, heads and layers. Higher attention density represents more attention allocation between tokens, while lower one means more focused and sparser attention.

**Sink ratio (sink).** The sink ratio measures the mean attention weight that all queries assign to the first token, i.e., the sink token. It reports how serious the attention sink phenomenon. For standard Transformer using softmax function, the sink ratio and the attention density sums one.

## C.4 BENCHMARKS

Table 3: The statistics of the benchmarks used in the overall experiment.

| Dataset | Sample Size |
|---|---|
| Wikitext | 60,634 |
| Lambada | 60,000 |
| PIQA | 16,113 |
| Hellaswag | 70,000 |
| WinoGrande | 44,000 |
| ARC | 7,787 (Easy Set + Challenge Set) |
| SIQA | 15,554 |
| BoolQ | 15,942 |

For our overall experiment, we compare models on eight common-sense reasoning tasks, in Table 3:

**Wikitext** (Merity et al., 2017): A large linguistic corpus extracted from Wikipedia articles, containing over 100 million word tokens. It tests a model's ability to predict the next word in a passage of text.

**Lambada** (Paperno et al., 2016): The LAmBdA dataset tests a model's capability of using broad discourse context to predict the last word of a passage extracted from books. It contains over 60,000 examples.

**PIQA** (Bisk et al., 2020): The Physical Interaction: Question Answering (PIQA) dataset tests commonsense reasoning about physical interactions between two entities. It contains 16,113 multiple choice questions generated from crowd-sourcing.

**Hellaswag** (Zellers et al., 2019): The HellaSwag dataset consists of 70,000 multiple choice questions about inferring what might happen next in a story. It requires commonsense reasoning to choose the most plausible ending.

**WinoGrande** (Sakaguchi et al., 2021): The WinoGrande dataset tests coreference resolution and commonsense reasoning with 44,000 examples obtained from books and websites.

**ARC** (Clark et al., 2018): The AI2 Reasoning Challenge (ARC) dataset contains 7,787 genuine grade-school level, multiple-choice science questions, grouped into an Easy Set (ARC-e) and a Challenge Set (ARC-c).

**SIQA** (Sap et al., 2019): The Social Interaction QA (SIQA) dataset contains 15,554 multiple choice questions that describe situations about people's social interactions.

**BoolQ** (Clark et al., 2019): The Boolean Questions (BoolQ) dataset contains 15,942 English yes/no questions sampled from Google search queries to test a model's ability to answer simple questions.

## D    LEARNED ATTENTION BIAS

As demonstrated in Figure 8, the learnable attention biases exhibit effective values within a range of approximately 100 tokens. When pre-training models with RoPE or ALiBi, which inherently impose strong long-range decay, the learnable range becom es further restricted to within 20 tokens. Beyond this range, the severely diminished attention weights by RoPE prevent the model from learning meaningful positional biases, resulting instead in periodic fluctuations without useful patterns, as shown in Figure 9. This also explains why ALiBi slopes show negligible changes when trained from standard initialization, because the initial long-range decay is so pronounced that it prevents the model from learning useful positional information.

## E    ELASTIC SOFTMAX

To better understand the behavior of Elastic-Softmax, we visualize all learnable offsets $\tau^{(h)}$ across layers and heads. In Fig. 10, the vertical axis indexes Transformer layers (bottom to top), and the horizontal axis enumerates the attention heads within each layer. Each cell corresponds to a head–layer pair, with color intensity representing the magnitude of the learned offset.

**Early layers**    Offsets are generally large, producing aggressive suppression of weak attention scores. This enforces sparse local focus, anchoring each query to nearby tokens and preventing early layers from being dominated by global noise or sink concentration at the sequence start. In effect, these layers act as filters that stabilize shallow representations.

**Middle layers**    Offsets decrease to moderate values. This allows informative cross-token connections to form. Within each layer, different heads learn distinct thresholds: some sustain high values to enforce conservative structural focus, while others adopt lower values that facilitate phrase-level or mid-range integration. This head-wise heterogeneity indicates a functional division of labor.

**Upper layers**    Offsets exhibit increased head-wise variance rather than uniformly small values. Suppression becomes heterogeneous: some heads keep higher thresholds to preserve sparse, high-precision links, while others lower thresholds to admit context. This stage functions as a final refinement that selectively emphasizes the most relevant tokens and avoids sink-like redistribution.

Overall, the learned offsets follow a hierarchical focusing pattern: aggressive pruning in the lower layers, stable selective filtering in the middle, and diverse refinement in the upper layers. Through the subtraction—ReLU mechanism, Elastic-Softmax achieves true sparsification by eliminating negligible weights while preventing sink-like accumulation, thereby maintaining a balanced trade-off between coverage and focus that improves both representational stability and predictive accuracy.

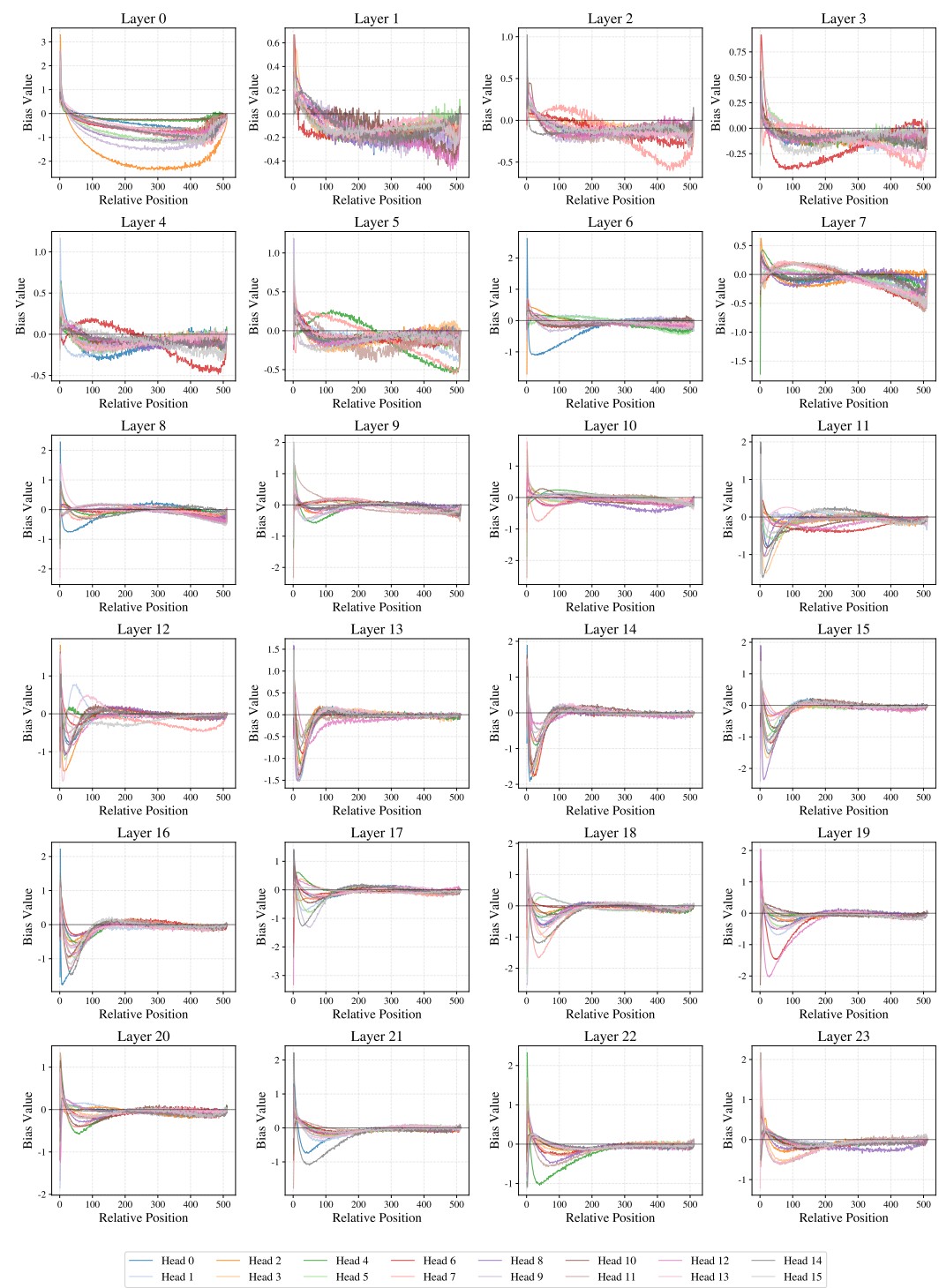

Figure 8: Learned position-dependent attention biases across all layers and attention heads. The max range of the bias is 1024, which is the same as pre-training length.

## F    COMPATIBILITY WITH FLASHATTENTION

Our Elastic-Softmax remains compatible with the FlashAttention framework, but requires two sequential passes due to the elastic filtering.

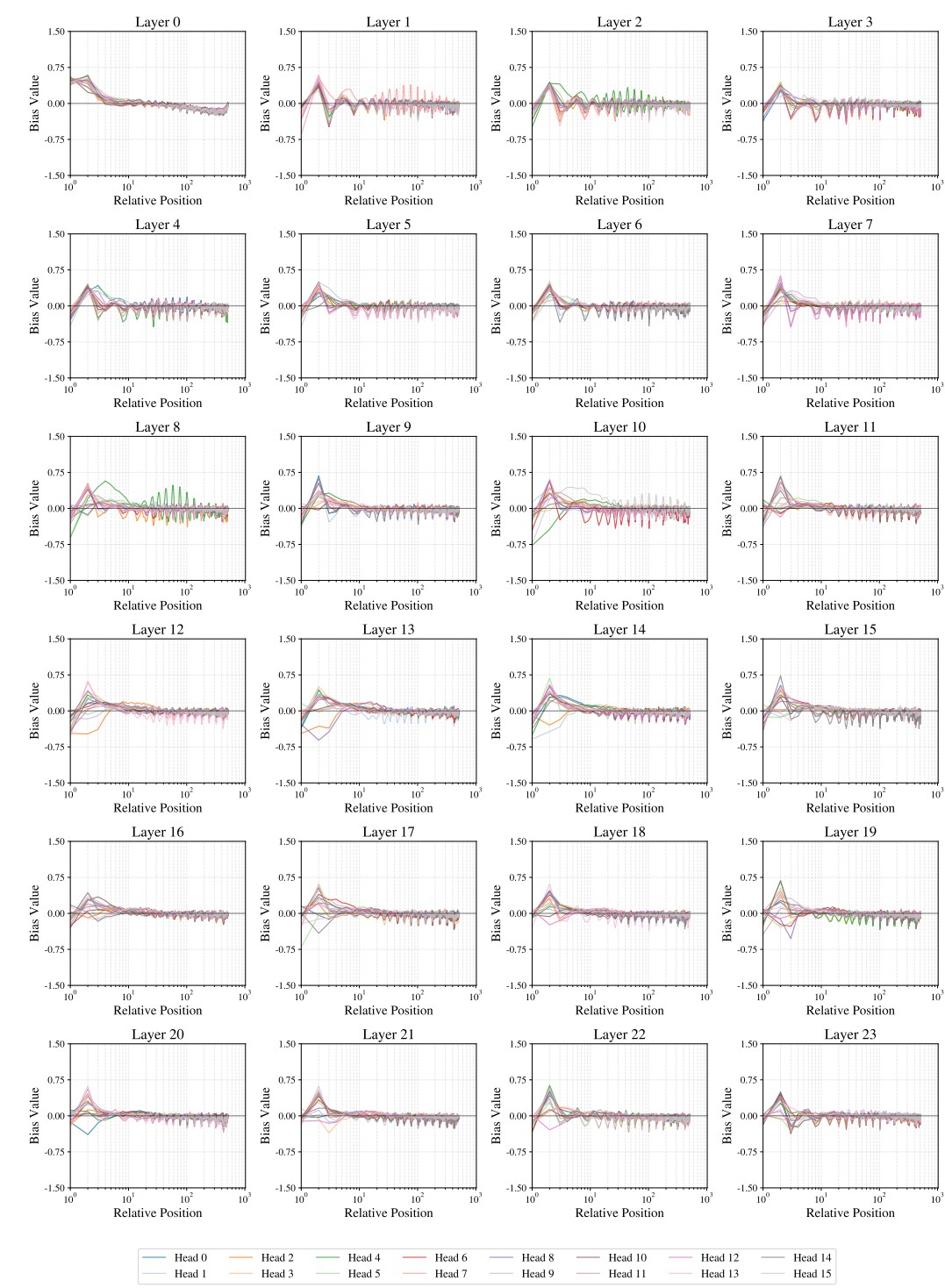

Figure 9: Learned position-dependent attention biases across all layers and attention heads with RoPE. The max range of the bias is 512, while the pre-training length is 1024. The

**Pass 1: Softmax Statistics**  For each query position $i$ under head $h$, compute the numerically stable softmax weights:

$$\tilde{\alpha}_{ij}^{(h)} = \frac{\exp(s_{ij}^{(h)} - m_i)}{\sum_{k=1}^{i} \exp(s_{ik}^{(h)} - m_i)}, \quad m_i = \max_{1 \le k \le i} s_{ik}^{(h)}. \tag{6}$$

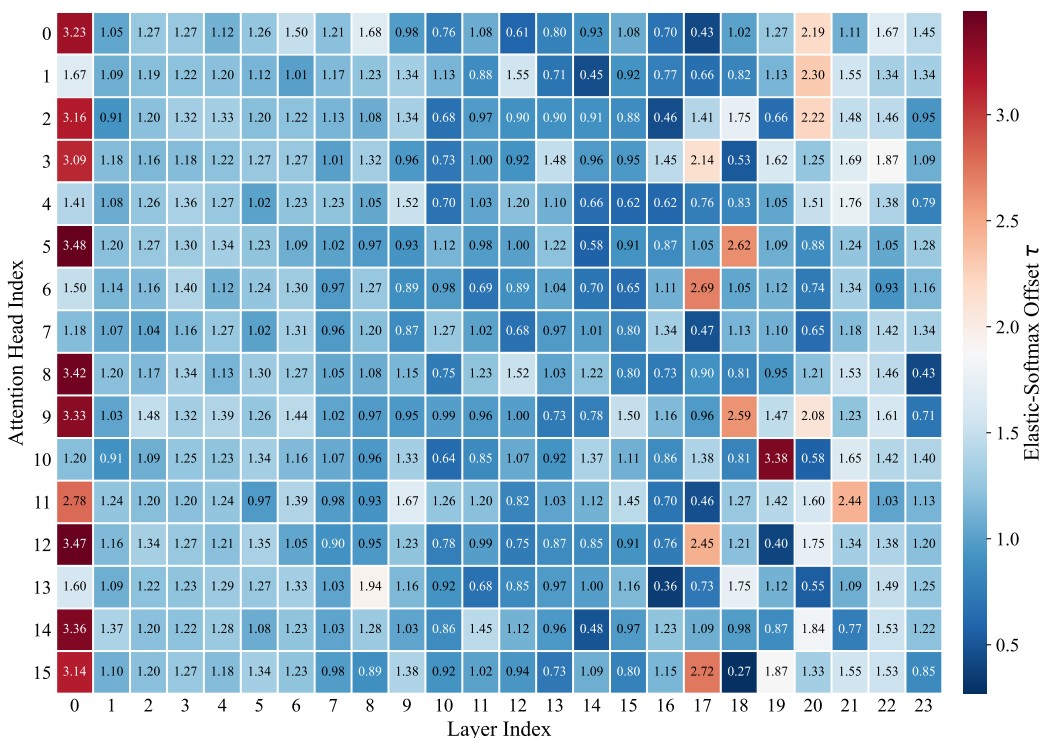

Figure 10: The demonstration of the learnable threshold of each attention head in each layer.

This step is identical to the first stage of FlashAttention and only requires computing the max and normalization statistics. No filtering is applied yet.

**Pass 2: Elastic Process**  Apply the learnable offset $\tau^{(h)}$ and ReLU filter to obtain sparse attention:

$$\alpha_{ij}^{(h)} = \text{ReLU}\left(\tilde{\alpha}_{ij}^{(h)} - \frac{\tau^{(h)}}{i}\right), \quad \text{Attn}_i^{(h)} = \sum_{j=1}^{i} \alpha_{ij}^{(h)} v_j^{(h)}. \tag{7}$$

Here, the offset is distributed evenly across $i$ candidate tokens, while the ReLU ensures non-negativity of the resulting weights, effectively setting suppressed positions to zero. This second pass involves only element-wise operations and a weighted sum, which can be fused as the FlashAttention kernel with minimal overhead.

**Summary.**  Although Elastic-Softmax requires two forward passes, both steps are linear in sequence length and reuse the same memory-efficient data layout as FlashAttention. Thus, the overall complexity remains $O(n^2)$ in time and $O(n)$ in memory.

## G  STATISTICAL PROPERTIES OF THE SINK TOKEN

**Setup.**  To examine how the sink signal is carried through the transformer layers, we measure the statistics of hidden states and Q/K/V vectors under two input formats: (i) natural text and (ii) repeated tokens. We report results on Qwen3-4B and BLOOM with different RPEs, RoPE, and ALiBi (Tables 4–7), respectively.

**Key observation 1: a low-variance footprint in representations.**  With natural text, token representations are heterogeneous: the norms and variances of Q/K/V vectors and hidden-states vary across positions, and spread broadly (Qwen3-4B: Tables 4a, 5a; BLOOM: Tables 6a, 7a). In contrast, the *sink token* exhibits a distinctive representational pattern: its hidden-state variance and the

Table 4: Statistics of Q/K/V attention vectors in Qwen3-4B.

(a) Non-repeated tokens

| Idx | ID | Token | Q-Norm | K-Norm | V-Norm | Q-Var | K-Var | V-Var |
|---|---|---|---|---|---|---|---|---|
| 0 | 12522 | Once | 1.4618 | 3.0660 | 0.4614 | 0.0168 | 0.0717 | 0.0017 |
| 1 | 5193 | upon | 6.8848 | 5.0653 | 4.1719 | 0.3720 | 0.2013 | 0.1308 |
| 2 | 264 | a | 7.2329 | 4.5908 | 2.6657 | 0.4118 | 0.1659 | 0.0547 |
| 3 | 882 | time | 6.3941 | 4.7750 | 3.6258 | 0.3216 | 0.1780 | 0.1007 |
| 4 | 11 | , | 6.8703 | 4.3678 | 2.6090 | 0.3691 | 0.1490 | 0.0532 |
| 5 | 304 | in | 7.2622 | 4.9094 | 4.3038 | 0.4151 | 0.1890 | 0.1337 |
| 6 | 264 | a | 6.3089 | 5.2083 | 4.4015 | 0.3134 | 0.2135 | 0.1433 |
| 7 | 4268 | land | 6.8214 | 5.4686 | 4.7506 | 0.3645 | 0.2335 | 0.1756 |
| 8 | 3041 | far | 8.2036 | 5.4945 | 4.5663 | 0.5299 | 0.2362 | 0.1612 |
| 9 | 11 | , | 7.7383 | 4.8891 | 3.6458 | 0.4689 | 0.1842 | 0.1042 |
| 10 | 3041 | far | 7.3633 | 5.5350 | 4.3791 | 0.4269 | 0.2389 | 0.1505 |
| 11 | 3123 | away | 7.8570 | 5.7813 | 4.5723 | 0.4830 | 0.2509 | 0.1620 |
| 12 | 11 | , | 7.1489 | 5.2523 | 4.0587 | 0.4024 | 0.2124 | 0.1293 |
| 13 | 1052 | there | 6.0721 | 4.3836 | 2.8356 | 0.2896 | 0.1513 | 0.0606 |
| 14 | 12163 | lived | 6.8271 | 5.4724 | 4.6806 | 0.3670 | 0.2345 | 0.1724 |

(b) Repeated token "sink"

| Idx | ID | Token | Q-Norm | K-Norm | V-Norm | Q-Var | K-Var | V-Var |
|---|---|---|---|---|---|---|---|---|
| 0 | 19309 | sink | 1.4635 | 3.0531 | 0.4627 | 0.0168 | 0.0711 | 0.0017 |
| 1 | 19309 | sink | 1.4635 | 3.0531 | 0.4627 | 0.0168 | 0.0711 | 0.0017 |
| 2 | 19309 | sink | 1.4635 | 3.0531 | 0.4627 | 0.0168 | 0.0711 | 0.0017 |
| 3 | 19309 | sink | 1.4636 | 3.0531 | 0.4627 | 0.0169 | 0.0711 | 0.0017 |
| 4 | 19309 | sink | 1.4636 | 3.0531 | 0.4608 | 0.0168 | 0.0711 | 0.0017 |
| 5 | 19309 | sink | 1.4646 | 3.0531 | 0.4608 | 0.0169 | 0.0711 | 0.0017 |
| 6 | 19309 | sink | 1.4646 | 3.0531 | 0.4608 | 0.0169 | 0.0711 | 0.0017 |
| 7 | 19309 | sink | 1.4635 | 3.0531 | 0.4608 | 0.0168 | 0.0711 | 0.0017 |
| 8 | 19309 | sink | 1.4620 | 3.0522 | 0.4607 | 0.0168 | 0.0711 | 0.0017 |
| 9 | 19309 | sink | 1.4642 | 3.0531 | 0.4608 | 0.0169 | 0.0711 | 0.0017 |
| 10 | 19309 | sink | 1.4631 | 3.0660 | 0.4627 | 0.0168 | 0.0717 | 0.0017 |
| 11 | 19309 | sink | 1.4646 | 3.0531 | 0.4608 | 0.0169 | 0.0711 | 0.0017 |
| 12 | 19309 | sink | 1.4636 | 3.0531 | 0.4627 | 0.0168 | 0.0711 | 0.0017 |
| 13 | 19309 | sink | 1.4634 | 3.0660 | 0.4627 | 0.0168 | 0.0717 | 0.0017 |
| 14 | 19309 | sink | 1.4635 | 3.0531 | 0.4627 | 0.0168 | 0.0711 | 0.0017 |

Table 5: Statistics of hidden states norms and variances in Qwen3-4B.

(a) Non-repeated tokens

| dx | ID | Token | Norm | Var |
|---|---|---|---|---|
| 0 | 12522 | Once | 7.1974 | 0.0202 |
| 1 | 5193 | upon | 16.5617 | 0.1071 |
| 2 | 264 | a | 15.0514 | 0.0885 |
| 3 | 882 | time | 16.9322 | 0.1120 |
| 4 | 11 | , | 14.6386 | 0.0837 |
| 5 | 304 | in | 16.5990 | 0.1076 |
| 6 | 264 | a | 15.3230 | 0.0917 |
| 7 | 4268 | land | 16.3703 | 0.1047 |
| 8 | 3041 | far | 16.5625 | 0.1072 |
| 9 | 11 | , | 16.9450 | 0.1121 |
| 10 | 3041 | far | 17.3868 | 0.1181 |
| 11 | 3123 | away | 17.6512 | 0.1217 |
| 12 | 11 | , | 16.5142 | 0.1065 |
| 13 | 1052 | there | 14.1547 | 0.0782 |
| 14 | 12163 | lived | 16.1331 | 0.1017 |

(b) Repeated token "sink"

| Idx | ID | Token | Norm | Var |
|---|---|---|---|---|
| 0 | 19309 | sink | 7.1961 | 0.0202 |
| 1 | 19309 | sink | 7.1961 | 0.0202 |
| 2 | 19309 | sink | 7.1961 | 0.0202 |
| 3 | 19309 | sink | 7.1961 | 0.0202 |
| 4 | 19309 | sink | 7.1960 | 0.0202 |
| 5 | 19309 | sink | 7.1959 | 0.0202 |
| 6 | 19309 | sink | 7.1959 | 0.0202 |
| 7 | 19309 | sink | 7.1961 | 0.0202 |
| 8 | 19309 | sink | 7.1906 | 0.0202 |
| 9 | 19309 | sink | 7.1959 | 0.0202 |
| 10 | 19309 | sink | 7.1962 | 0.0202 |
| 11 | 19309 | sink | 7.1959 | 0.0202 |
| 12 | 19309 | sink | 7.1961 | 0.0202 |
| 13 | 19309 | sink | 7.1962 | 0.0202 |
| 14 | 19309 | sink | 7.1961 | 0.0202 |

variance of its *value* vectors are markedly smaller than those of neighboring tokens (Qwen3-4B: Tables 4b, 5b; BLOOM: Tables 6b, 7b). This indicates that the sink effect propagates through the model via a low-variance state that is easy to recognize.

**Key observation 2: repetition collapses positional separability.** Under repeated-token inputs, *all* summary statistics flatten out: Q/K/V norms and variances, as well as hidden-state norms and variances, become nearly uniform across positions (Qwen3-4B: Tables 4b, 5b; BLOOM: Tables 6b, 7b). In effect, every position behaves like a sink token once semantic differences are removed, indicating that the model can no longer disambiguate identical tokens by their representations alone.

**Implication for RPEs (RoPE/ALiBi).** This collapse appears under both RoPE and ALiBi, which suggests that relative position encodings chiefly act by shaping *attention scores* (i.e., injecting position-sensitive biases) rather than leaving a persistent positional information in the hidden states. Consequently, representation-level positional separability is absent when inputs repeat, and sink-like dynamics emerge at all positions. These findings motivate our design to (i) strengthen positional discrimination at the scoring level via RoPE combined with learnable, head-wise distance biases, and (ii) suppress sink-driven background attention weight with Elastic-Softmax.

Table 6: Statistics of Q/K/V attention vectors in BLOOM.

(a) Non-repeated tokens

| Idx | ID | Token | Q-Norm | K-Norm | V-Norm | Q-Var | K-Var | V-Var |
|---|---|---|---|---|---|---|---|---|
| 0 | 64393 | Once | 6.1174 | 12.9978 | 0.1502 | 0.5761 | 2.6465 | 0.0004 |
| 1 | 14591 | upon | 9.0732 | 15.7411 | 6.0661 | 1.2908 | 3.6578 | 0.5768 |
| 2 | 267 | a | 7.4769 | 13.8969 | 5.2563 | 0.8773 | 2.9292 | 0.4306 |
| 3 | 3509 | time | 8.2724 | 14.0432 | 6.4744 | 1.0788 | 2.9424 | 0.6650 |
| 4 | 15 | , | 8.0348 | 13.8262 | 6.9108 | 1.0223 | 2.8416 | 0.7457 |
| 5 | 361 | in | 8.3171 | 13.5464 | 5.7878 | 1.0759 | 2.6934 | 0.5292 |
| 6 | 267 | a | 8.5494 | 13.6764 | 5.2263 | 1.0866 | 2.8268 | 0.4335 |
| 7 | 11970 | land | 8.4330 | 14.4047 | 4.9709 | 1.1134 | 3.1245 | 0.3856 |
| 8 | 8372 | far | 9.0026 | 14.4584 | 6.3974 | 1.2844 | 3.0141 | 0.6492 |
| 9 | 15 | , | 9.1776 | 14.2987 | 6.6702 | 1.3166 | 2.8046 | 0.6983 |
| 10 | 8372 | far | 9.2586 | 15.1883 | 8.7518 | 1.3578 | 3.2382 | 1.1968 |
| 11 | 14723 | away | 7.7820 | 15.4436 | 6.8175 | 0.9355 | 3.5861 | 0.7086 |
| 12 | 15 | , | 7.6954 | 14.5350 | 6.7992 | 0.9393 | 3.2038 | 0.7161 |
| 13 | 2782 | there | 8.5690 | 14.6540 | 6.2535 | 1.1644 | 3.2433 | 0.6092 |
| 14 | 65532 | lived | 8.3138 | 13.9679 | 5.7587 | 1.0638 | 2.9913 | 0.4716 |

(b) Repeated tokens

| Idx | ID | Token | Q-Norm | K-Norm | V-Norm | Q-Var | K-Var | V-Var |
|---|---|---|---|---|---|---|---|---|
| 0 | 66037 | sink | 6.1010 | 12.9761 | 0.1262 | 0.5716 | 2.6375 | 0.0002 |
| 1 | 66037 | sink | 6.1010 | 12.9761 | 0.1262 | 0.5716 | 2.6375 | 0.0002 |
| 2 | 66037 | sink | 6.1010 | 12.9761 | 0.1262 | 0.5716 | 2.6375 | 0.0002 |
| 3 | 66037 | sink | 6.1010 | 12.9761 | 0.1263 | 0.5716 | 2.6375 | 0.0002 |
| 4 | 66037 | sink | 6.1011 | 12.9760 | 0.1263 | 0.5716 | 2.6374 | 0.0002 |
| 5 | 66037 | sink | 6.1011 | 12.9761 | 0.1263 | 0.5716 | 2.6375 | 0.0002 |
| 6 | 66037 | sink | 6.1010 | 12.9761 | 0.1263 | 0.5716 | 2.6375 | 0.0002 |
| 7 | 66037 | sink | 6.1010 | 12.9760 | 0.1263 | 0.5716 | 2.6374 | 0.0002 |
| 8 | 66037 | sink | 6.1010 | 12.9760 | 0.1263 | 0.5716 | 2.6374 | 0.0002 |
| 9 | 66037 | sink | 6.1010 | 12.9761 | 0.1262 | 0.5716 | 2.6375 | 0.0002 |
| 10 | 66037 | sink | 6.1010 | 12.9761 | 0.1262 | 0.5716 | 2.6374 | 0.0002 |
| 11 | 66037 | sink | 6.1010 | 12.9760 | 0.1263 | 0.5716 | 2.6374 | 0.0002 |
| 12 | 66037 | sink | 6.1011 | 12.9760 | 0.1262 | 0.5716 | 2.6374 | 0.0002 |
| 13 | 66037 | sink | 6.1010 | 12.9761 | 0.1262 | 0.5716 | 2.6375 | 0.0002 |
| 14 | 66037 | sink | 6.1010 | 12.9760 | 0.1263 | 0.5716 | 2.6374 | 0.0002 |

Table 7: Statistics of hidden-state in BLOOM.

(a) Non-repeated tokens

| Idx | ID | Token | Norm | Var |
|---|---|---|---|---|
| 0 | 64393 | Once | 37.6291 | 1.3823 |
| 1 | 14591 | upon | 46.2054 | 2.0837 |
| 2 | 267 | a | 43.6723 | 1.8613 |
| 3 | 3509 | time | 46.7614 | 2.1333 |
| 4 | 15 | , | 42.5041 | 1.7633 |
| 5 | 361 | in | 42.2835 | 1.7447 |
| 6 | 267 | a | 42.9327 | 1.7988 |
| 7 | 11970 | land | 44.4018 | 1.9240 |
| 8 | 8372 | far | 45.2311 | 1.9961 |
| 9 | 15 | , | 46.5993 | 2.1190 |
| 10 | 8372 | far | 48.5840 | 2.3038 |
| 11 | 14723 | away | 44.8350 | 1.9613 |
| 12 | 15 | , | 39.5864 | 1.5297 |
| 13 | 2782 | there | 44.0638 | 1.8954 |
| 14 | 65532 | lived | 44.2725 | 1.9125 |

(b) Repeated token "sink"

| Idx | ID | Token | Norm | Var |
|---|---|---|---|---|
| 0 | 66037 | sink | 37.6223 | 1.3818 |
| 1 | 66037 | sink | 37.6223 | 1.3818 |
| 2 | 66037 | sink | 37.6223 | 1.3818 |
| 3 | 66037 | sink | 37.6223 | 1.3818 |
| 4 | 66037 | sink | 37.6223 | 1.3818 |
| 5 | 66037 | sink | 37.6223 | 1.3818 |
| 6 | 66037 | sink | 37.6223 | 1.3818 |
| 7 | 66037 | sink | 37.6223 | 1.3818 |
| 8 | 66037 | sink | 37.6223 | 1.3818 |
| 9 | 66037 | sink | 37.6223 | 1.3818 |
| 10 | 66037 | sink | 37.6223 | 1.3818 |
| 11 | 66037 | sink | 37.6223 | 1.3818 |
| 12 | 66037 | sink | 37.6223 | 1.3818 |
| 13 | 66037 | sink | 37.6223 | 1.3818 |
| 14 | 66037 | sink | 37.6223 | 1.3818 |

