# OpenReview forum: "Attention Needs to Focus: A Unified Perspective on Attention Allocation"
_ICLR.cc/2026/Conference — Submitted to ICLR 2026_

### Official Review · Reviewer_Ayxf · 2025-10-29

**Soundness:** 2
**Presentation:** 3
**Contribution:** 3
**Rating:** 4
**Confidence:** 4

**Summary:**

This paper identifies two fundamental failure modes in the standard Transformer self-attention mechanism: Attention Overload (attention is spread too broadly, blurring semantic features and causing representational collapse) and Attention Underload (attention is forced onto irrelevant tokens like "attention sinks" due to softmax's normalization constraint).

To address both issues within a unified framework, the authors propose Lazy Attention, which consists of two key components:

**Positional Discrimination**: A hybrid positional encoding combining RoPE with learnable, head-specific attention biases to sharpen token distinctions and alleviate overload.

**Elastic-Softmax**: A modified normalization that subtracts a learnable, head-specific offset before applying a ReLU, allowing the model to assign zero attention to irrelevant tokens and mitigate underload.

**Strengths:**

**Unified Perspective**: The paper proposes a unified theoretical framework that connects two seemingly disparate problems (representational collapse and attention sink) to a single root cause: improper attention allocation.

**Comprehensive Evaluation**: The method is rigorously tested against a wide array of strong baselines, including Transformer variants, recurrent architectures (Mamba2, RetNet), and streaming inference methods, across multiple model scales and nine diverse benchmarks.

**Compatibility**: The authors explicitly address and ensure the method's compatibility with optimized kernels like FlashAttention, a crucial point for real-world adoption.

**Weaknesses:**

**Limited Scale of Models**: The largest model scale tested is 760M parameters. While this is standard for architectural research, the effectiveness of Lazy Attention in today's state-of-the-art models (e.g., those with 10B+ parameters) remains an open question. The behavior of attention mechanisms can change significantly with scale.

**Computational Overhead of Elastic-Softmax**: The paper notes that Elastic-Softmax requires two passes to be compatible with FlashAttention. While the complexity remains O(n²), the actual wall-clock time and memory overhead compared to a single-pass softmax are not quantified. This is an important practical consideration.

**Ablation on Sparsity's Utility**: While the high sparsity is celebrated, the paper does not directly demonstrate the resulting computational speedups or memory savings during inference. Showing end-to-end latency improvements would strengthen the claim of efficiency.

**Questions:**

1. How do the authors anticipate the performance and the learned behaviors (e.g., bias patterns, offset values) of Lazy Attention would change when scaled to models with billions of parameters? Do you expect the "overload" and "underload" phenomena to persist or change in nature at that scale?

2. The authors mention Elastic-Softmax is compatible with FlashAttention but requires two passes. Could you quantify the resulting training and inference latency overhead compared to standard softmax attention? Does the achieved sparsity translate into a net reduction in inference time?

3. The initialization of the Elastic-Softmax offset τ=1.0 is crucial. Was this value found through extensive ablation? What is the sensitivity of the model's final performance to this initial value?

4. How does Lazy Attention, which learns sparsity dynamically, compare conceptually and empirically to pre-defined sparse attention patterns (e.g., sliding window, BigBird)? Does it learn similar patterns, or does it discover more optimal, task-specific sparsity?

---

### Official Review · Reviewer_q3b5 · 2025-11-03

**Soundness:** 3
**Presentation:** 3
**Contribution:** 3
**Rating:** 6
**Confidence:** 3

**Summary:**

The paper argues that two pervasive Transformer pathologies—representational collapse and attention sink—stem from the same root cause: improper attention allocation. It diagnoses two extremes: Attention Overload (too many tokens get comparably high weights) and Attention Underload (no token is relevant, but softmax must distribute probability mass somewhere, often to early “sink” tokens). To address both, the authors propose Lazy Attention, which combines: 1) Positional Discrimination: standard RoPE plus learnable distance‑dependent, head‑wise biases added to the score to sharpen distinctions across heads and dimensions. Heatmaps show varied learned decay patterns across layers/heads. 2) Elastic‑Softmax: a post‑softmax, per‑head offset with ReLU, to zero‑out weights of irrelevant keys and neutralize sink behavior. The authors visualize learned offsets by layer  and provide alternatives/ablations.

On FineWeb‑Edu pretraining (10B/100B tokens) with 340M and 760M models, Lazy Attention achieves competitive or better accuracy on 8 reasoning benchmarks while reducing sink mass and increasing sparsity. The method also shows improved length extrapolation from a 512‑token training context to 1024/2048 on WikiText/LAMBADA. For implementation, Elastic‑Softmax is described as FlashAttention‑compatible via two passes.

**Strengths:**

++ The paper connects collapse and sink via “allocation extremes,” then uses interventions to localize and characterize sink behavior and its dependence on positional encoding. This yields two crisp takeaways about variance footprints and the role of RPEs in shaping weights rather than embeddings.

++ The learnable head‑wise distance biases plus RoPE sharpen focus; the Elastic‑Softmax filter removes low‑relevance mass and alleviates sink. The layer‑wise offset patterns and bias curves are informative and align with the overload/underload story.

++ At 340M/760M, accuracy is on par or better than strong baselines across diverse tasks while showing 59.58% sparsity and reduced sink ratio. Perplexity degrades less with increasing length from 512→2048 versus a standard Transformer.

**Weaknesses:**

-- The paper defines 𝛼=ReLU(softmax−𝜏_𝑖) and does not re‑normalize before applying 𝑉, so the sum of weights is no longer 1, making the output scale data‑/offset‑dependent. The metric density/sink is clear for softmax (sum=1) but less interpretable when mass is removed. In addition, there are sign/initialization inconsistencies: §4.2 sets 𝜏0^(ℎ)=1 and “divided evenly across the 𝑖 attended tokens,” whereas Table 2 reports the best variant using 𝜏_ℎ/seq_len; Fig. 5 shows negative offsets in early layers yet claims “aggressive filtering,” which contradicts the subtract‑then‑ReLU formula. These details impact stability and should be clarified with formulas and pseudocode.

-- Elastic‑Softmax requires two passes (App. F), and head‑wise distance biases ​introduce many parameters (Fig. 8/9). The paper lacks wall‑clock training/inference time, throughput/latency, and KV‑cache or peak memory comparisons versus standard attention and other sparse/streaming variants. Without these, the claimed practical benefits (e.g., sparsity) are hard to evaluate.

-- The method “adopts a larger RoPE base 𝐵” but does not specify the value/rationale nor ablate it. Likewise, the distance‑bias parameterization (range, bucketization vs. per‑distance tables, regularization) is not detailed; Fig. 8 caption mentions a max range 1024, conflicting with §5.1’s 4096 context length. Clear specs and ablations on 𝐵, bias ranges, and learned patterns would bolster the claims.

**Questions:**

1. Exact Elastic‑Softmax algorithm: Do you re‑normalize after thresholding? If not, how do you prevent scale drift across layers/heads?

2. Compute/memory impact: What are the measured FWD/BWD times and max memory for Lazy Attention vs. softmax attention on the same hardware and sequence lengths, including the two‑pass overhead? Please include KV‑cache and activation footprints.

3. RoPE base 𝐵 & bias parameterization: What 𝐵 did you use, and how sensitive are results to 𝐵? Are the distance biases bucketed (like T5/ALiBi) or per distance up to a cutoff (Fig. 8/9)? How big is the parameter table per head/layer, and how does it scale to 32k+ tokens?

4. Normalization‑aware metrics: Since Elastic‑Softmax can reduce total mass, how are density and sink computed? Could you report (i) total surviving mass, (ii) fraction assigned to sink, and (iii) fraction to non‑sink, so the numbers remain interpretable across methods?

5. Very‑long and streaming settings: How does Lazy Attention perform on >32k context and standard long‑context QA (e.g., LongBench variants)? Any interaction with streaming methods (e.g., StreamingLLM) once sink is suppressed?

6. Gradients through ReLU thresholding: Early in training, many weights may be zeroed; did you observe optimization instabilities or head collapse? Any tricks (warmups/entropy terms) to keep gradients flowing?

---

### Official Review · Reviewer_Wg8F · 2025-11-03

**Soundness:** 2
**Presentation:** 2
**Contribution:** 2
**Rating:** 2
**Confidence:** 4

**Summary:**

This paper identifies that the representational collapse and attention sink in LLMs stem from improper attention allocation, which manifests as two failure modes: Attention Overload and Attention Underload. To address these issues, it proposes Lazy Attention, integrating Positional Discrimination (combining RoPE with learnable attention biases) and Elastic-Softmax (a modified normalization function).

**Strengths:**

1. The paper provides a unified perspective to explain two long-standing problems (representational collapse and attention sink) in Transformer architectures,.
2. Extensive experiments on diverse benchmarks and different model scales demonstrate that Lazy Attention not only mitigates attention sink but also achieves competitive performance compared to state-of-the-art baselines in some cases.

**Weaknesses:**

1.The paper lacks clear definitions and explanations for key elements in Figure 2. For Figure 2a, the notation "Mask@2" is introduced when inserting a fixed [Mask] token during pre-training, but it fails to specify what "2" refers to.

2. In Line 254, the notations "d" and "D" are not explicitly defined.

3. Formula 4 introduces a learnable attention bias term b^(h)_{|i-j|}, which is a head-specific bias dependent on the relative distance |i-j| between tokens. The paper does not discuss the potential memory overhead of this bias: for each attention head, a bias value needs to be stored for every possible relative distance (up to the maximum sequence length, e.g., 4096 in experiments). As the sequence length increases, the number of bias parameters grows linearly with the sequence length, which may significantly increase memory consumption, especially for long-context models. Additionally, the paper does not evaluate whether this distance-dependent learnable bias affects the model’s extrapolation ability—whether the bias learned for short sequences can generalize to longer sequences remains unaddressed.

4. The results in Table 1 show an inconsistent pattern between WikiText perplexity (ppl) and LAMBADA (LMB) ppl for the proposed Lazy Attention, which raises doubts about its effectiveness. For the 340M parameter model, Lazy Attention achieves a WikiText ppl of 25.32, but its LMB ppl is 31.84, which is worse than baselines. The paper does not explain this inconsistency.

5. There are inconsistencies in the notation of the Elastic-Softmax offset. In Formula 4 (Section 4.1), the paper uses the notation $\tau_0^h$ for the offset, while in Table 2, the notation is written as $\tau_h^0$. It is not clarified whether these are two distinct notations for the same parameter. Furthermore, Formula 4 implies that the default initial value of $\tau_0^h$ is 0, but Table 2 shows that the variant with $\tau_0^h$ = 1 achieves better performance.

6. Figure 6 compares the perplexity of different models across varying sequence lengthsfor length extrapolation, but the superiority/inferiority of Lazy Attention relative to baselines is inconsistent between subfigure (a) and subfigure (b).

7. In, Formula 5, Elastic-Softmax relaxes the softmax constraint that attention weights sum to 1. The paper does not address the potential training instability caused by non-normalized attention weights: if the sum of attention weights is too large, it may amplify the impact of noisy tokens; if the sum is too small (e.g., most weights are suppressed to 0, leading to a sum close to 0), the model may fail to capture sufficient contextual information. The paper does not report whether such sum deviations occur during training.

8. The paper uses two different notations for learnable biases—"m" in Line 258 and "b" in Line 264—but does not clarify their differences.

**Questions:**

NA

---

### Official Review · Reviewer_PQ4b · 2025-11-07

**Soundness:** 3
**Presentation:** 3
**Contribution:** 3
**Rating:** 2
**Confidence:** 4

**Summary:**

The paper proposes Lazy Attention, which aims to unify two well-known attention pathologies (i.e., representational collapse and attention sink) under a single explanatory framework. The method combines Positional Discrimination (adding learnable head- and distance-specific biases on top of RoPE) and Elastic-Softmax (a ReLU-threshold applied after softmax) to promote sparse, focused attention. Experiments on 340M- and 760M-parameter models report moderate accuracy improvements, reduced sink metrics, and claimed attention sparsity of up to 59.6%.

**Strengths:**

- **Clear conceptual framing**. The paper presents a coherent narrative linking two common attention failure modes as manifestations of improper allocation, supported by intuitive visual analyses.

- **Simplicity and modularity**. Both proposed components, including score-level positional bias and post-softmax filtering, are easy to implement within standard transformer codebases.

- **Comprehensive qualitative evidence**. Attention heatmaps, offset distributions, and learned bias profiles provide interpretability and consistent visual trends.

**Weaknesses:**

- **Ambiguous core mechanism.** The paper defines Elastic-Softmax as applying a token-wise ReLU to `(softmax – τ_i^h)`, yet later sections describe thresholding as a fixed `τ_h / seq_len`. These two formulations are not equivalent and lead to different scaling behaviors. Moreover, in the reported ablations, the learned τ values often become negative, effectively *adding* probability mass instead of filtering low-attention entries, which is the opposite of the stated mechanism.

- **Unnormalized probability mass and scaling drift.** Applying a post-softmax ReLU breaks probability conservation, yet no renormalization is performed. As a result, attention magnitude may vary with sequence length or sparsity, confounding comparisons and potentially destabilizing training objectives.

- **Incomplete baseline coverage**. Strong probability-sparsifying or normalization-free alternatives (e.g., sparsemax, entmax) are omitted from large-scale comparisons, leaving the empirical novelty and significance of the proposed method under-substantiated.

- **Unmeasured real-device efficiency**. Despite claiming FlashAttention compatibility, the two-pass design lacks any measured latency/throughput/memory data or speedups tied to induced sparsity.

**Questions:**

What is the exact, implemented Elastic-Softmax formula (including threshold normalization and sign), and how does it reconcile with the reported offset statistics?

Do you renormalize the post-ReLU weights, and if so, how? If not, how do you control for scale drift across layers and sequence lengths?

Under which tasks/datasets does Elastic-Softmax (not merely the positional bias) yield statistically significant gains over its ablation across multiple seeds?

Can you add same-recipe comparisons against sigmoid attention and sparsemax/entmax at the larger scale and report variance, tuning budgets, and training stability?

---

### Meta-Review · Area_Chair_aaqr · 2025-12-29

**Summary:**

## Summary
This submission studies two known phenomena named representational collapse and attention sink in a unified framework and attributes the two phenomena as the “attention overload” and “attention underload,” respectively. Then the paper proposes Lazy Attention, which combines **positional discrimination** (which is essentially RoPE + learnable head-wise attention bias) and **elastic-softmax** (replacing softmax with ReLU(softmax - learnable bias)) to mitigate the phenomena. The authors validate the performance of the proposed method on several benchmarks.

## Reviewer Concerns
- **Clarity and readability issues**. Multiple questions were raised on the Elastic-Softmax mechanism, due to the confusing notation and inconsistent definition. Also, learned values of the offset $\tau$ are negative in some cases, which is the opposite of the intended purpose of Elastic-Softmax. More clarification questions regarding notation were raised by Reviewer Wg8F.
- **Potential instability by non-normalized attention**. A couple of reviewers pointed out that Elastic-Softmax may cause training instability because its attention weights no longer sum to one.
- **Lack of time/throughput/memory analysis**. Most reviewers raised the concern that the paper does not offer measurements of wall-clock time, throughput, and memory usage.
- **Memory requirement of learnable attention bias**. The learnable attention bias introduces parameters whose count is proportional to the sequence length and the number of heads.
- **Inconsistent performance improvement and missing baselines**. It was also pointed out that the improvement offered by the proposed method is not consistent over different models (e.g., LAMBADA perplexity is worse than baselines in Table 1 340M, w/o Elastic-softmax variant showing better length generalization performance in Figure 6). Reviewer PQ4b also pointed out that the paper doesn’t compare against other modifications to softmax such as sparsemax and entmax.

## Overall Assessment
Unfortunately, the authors did not respond to the reviews and the issues remain unresolved; therefore, I cannot recommend acceptance of this paper at this time.

**Reviewer Concerns:**

The authors did not submit their responses, hence all concerns remain unresolved.

**Reviewer Scores:**

The authors did not submit their responses, hence there would have been no change in the review scores.

---

### Decision · Program_Chairs · 2026-01-26

Reject